# Molecular Detection and Phylogenetic Characterization of *Anaplasma* spp. in Dogs from Hainan Province/Island, China

**DOI:** 10.3390/vetsci10050339

**Published:** 2023-05-10

**Authors:** Yang Lin, Sa Zhou, Archana Upadhyay, Jianguo Zhao, Chenghong Liao, Qingfeng Guan, Jinhua Wang, Qian Han

**Affiliations:** 1One Health Institute, Hainan University, Haikou 570228, China; 2College of Animal Science and Technology, Hainan University, Haikou 570228, China; 3Laboratory of Tropical Veterinary Medicine and Vector Biology, School of Life Sciences, Hainan University, Haikou 570228, China

**Keywords:** *Anaplasma*, dogs, Hainan, co-infection, prevalence

## Abstract

**Simple Summary:**

*Anaplasma* is a Gram-negative parasitic bacterium transmitted by ticks. It has become an important tick-borne pathogen that endangers human and animal health in recent years. Dogs exposed to ticks are more likely to test positive for *Anaplasma* infection. This study confirms that dogs in Hainan province/island, China, are exposed to *Anaplasma* and raises the possibility that people and other animals could contract the disease.

**Abstract:**

Anaplasmosis is a serious infection which is transmitted by ticks and mosquitos. There are very few reports and studies that have been carried out to understand the prevalence, distribution, and epidemiological profile of *Anaplasma* spp. infection in dogs in Hainan province/island. In the present study, we have tried to understand the prevalence, distribution, and occurrence of *Anaplasma* spp. infections in dogs (*n* = 1051) in Hainan Island/Province to establish a surveillance-based study. The confirmed positive samples by Polymerase chain reaction (PCR) were subjected to capillary sequencing for further strain-specific confirmation, followed by the construction of phylogenetic trees to determine their genetic relations. Various statistical tools were used to analyze related risk factors. There were three species of *Anaplasma* detected from the Hainan region; namely, *A. phagocytophilum*, *A. bovis*, and *A. platys*. The overall prevalence of *Anaplasma* is 9.7% (102/1051). *A. phagocytopihum* was prevalent in 1.0% of dogs (11/1051), *A. bovis* was found in 2.7% of dogs (28/1051), and *A. platys* in 6.0% of dogs (63/1051). Our surveillance-based study conducted to understand the occurrence and distribution pattern of *Anaplasma* spp. in Hainan will help in designing effective control measures along with management strategies so as to treat and control the infection in the area.

## 1. Introduction

*Anaplasma* spp. are important tick-borne bacteria distributed all over the world, and they are of great importance to veterinarians and public health. *Anaplasma* spp. can be characterized as Gram-negative bacteria that are transmitted by ticks, belonging to the family Anaplasmataceae, order Rickettsiales. As per the current characterizing system, the genus includes eight *Anaplasma* species (*A. phagocytophilum*, *A. marginale*, *A. centrale*, *A. ovis*, *A. bovis*, *A. platys*, *A. odocoilei*, and *A. capra*) and a large number of unclassified genovariants that have still not being assigned to any known species [1]. Humans and livestock are both prone to the infection caused by *Anaplasma* spp., giving rise to a broad range of clinical symptoms.

*A. phagocytophilum* has a broad host range and may cause severe complications in several mammalian species, including humans. A confirmed case of human granulocytic anaplasmosis was first found in the United States in 1994 [2]. In China, *A. phagocytophilum* occurred for the first time in Anhui Province in 2006, and it was found that the pathogen could potentially be transmitted through blood or respiratory secretions of HGA (human granulocytic anaplasmosis) patients [3]. *A. phagocytophilum* is the main causative agent of HGA [4]. The number of human anaplasmosis cases reported to Hainan Center for Disease Control has increased steadily since the disease became more prevalent, from 348 cases in the year 2000, to approximately 1800 cases in 2010. The incidence of anaplasmosis has also increased, from 1.4 cases per million persons in the year 2000 to 6.1 cases per million persons in 2010. However, the case fatality rate has still remained low at less than 1% [5,6,7].

*A. platys* is a parasite of canine platelets, and can cause infectious canine cyclic thrombocytopenia. In 1978, Harvey and his colleagues first described the disease in a canine infection in Florida, USA [8].

Anaplasmosis is caused by *A. bovis*, which presents various clinical symptoms, including fever, weight loss, and decreased milk production. In the acute phase of the disease, it might even lead to miscarriage and death in a few cases [9,10]. This pathogenic bacterium has been identified and found to be prevalent in China [11,12], Japan [13], Korea [14,15], and the Brazilian Pantanal [16].

Various risk factors such as the sex of the animal, herd management, seasons, tick presence, and herd size are reported to be associated with anaplasmosis. Spatio-temporal conditions such as vector habitat, bacterial populations, animal grazing systems, hygiene, and management practices also affect the epidemiology of the infection.

The main purpose of this study was not only to investigate the prevalence of *Anaplasma* spp. in dogs in Hainan, China, but also to unravel various management and strategic control mechanisms for the disease. This study will be fruitful in many ways to provide surveillance data which would, in turn help in the prevention and control of *Anaplasma* spp. and their infection in Hainan.

## 2. Materials and Methods

### 2.1. Study Site

Geographically, Hainan (also known as Hainan Island) is located in the South China Sea (between 108°37′ and 111°03′ E and 18°10′ and 20°10′ N). The island boasts a pleasant tropical climate, which makes it different from mainland China. It covers about 35,400 square kilometers of land, experiencing an average annual rainfall of 1000–2600 mm (mainly from July to October) and an average annual temperature of 26.5 °C.

### 2.2. Sampling

The study period was from June 2019 to December 2021 covering 18 cities/counties of Hainan Island/Province (Figure 1). Animal studies were approved by the Hainan University Institutional Animal Care and Use Committee. These locations are inhabited abundantly by tick populations, the *Anaplasma* spp. vectors. A number of samples collected in each city/county are shown in Table 1. A total of 1051 dogs were sampled. These comprised 20.6% (217/1051) of family-owned dogs from urban areas that were taken to veterinary clinics for routine health examinations, 27.8% (292/1051) of dogs from an animal rescue shelter (Animal Protection Association of Haikou), and 51.6% (542/1051) of farmer-owned dogs from villages in Hainan.

Approximately 2 mL of whole blood was collected from the lateral radial vein of each dog. Each blood sample was collected in the EDTA (anticoagulant) tube, and the collected samples were stored at −80 °C for further use. Additionally, in order to study the risk factors (age, sex, pesticide use, feeding type, tick infection) related to the pathogens of interest, a questionnaire was designed and collected during sampling. Each animal was physically examined to check for the presence of ticks, and they were divided in two groups on the basis of age: age < 1 year and age > 1 year.

### 2.3. Nucleic Acid Extraction

Genomic DNA was isolated from 100 μL of blood using the DNA Mini Kit (Sangon Biotech Co., Shanghai, China) according to the manufacturer’s instructions. The isolated DNA was stored at −20 °C for further downstream applications.

### 2.4. PCR Amplification

To confirm the presence of infection with the *Anaplasma* species, the nested PCR was carried out to amplify the 16S rRNA gene for *A. phagocytophilum* and *A. bovis*. The primer sequences and thermal cycling conditions used in this study are tabulated in Table 2 and Table 3. One-step PCR targeting the groEL gene was used to detect *A. platys*. For *A. phagocytophilum* and *A. bovis*, the first-round PCR reactions were performed in a final volume of 25 μL containing 12.5 μL 2 × Taq Plus Master Mix II (Dye Plus) (Sangon Biotech Co., Shanghai, China), 10.5 μL of nuclease-free water; 0.5 μL of each primer (EE1 and EE2 described by Barlough [17]); and 1 μL of genomic DNA as a template. The final volume of the second round of PCR reaction was 25 μL containing 12.5 μL 2 × Taq Plus Master Mix II (Dye Plus) (Sangon Biotech Co., Shanghai, China); 10.5 μL of nuclease-free water; 0.5 μL of each primer (SSAP2f and SSAP2r for *A. phagocytophilum*; AB1f and AB1r for *A. bovis* described by Kawahara [18]); and 1 μL of the amplified product from the first PCR reaction. The nested PCR reaction system has been tabulated in Table 4. For the amplification of *A. platys*, the partial segment of *gltA* gene was amplified using primers *gltAf* and *gltAr* [19]. The PCR reactions were performed in a final volume of 25 μL containing 12.5 μL 2 × Taq Plus Master Mix II (Dye Plus) (Sangon Biotech Co., Shanghai, China); 10.5 μL of nuclease-free water; 0.5 μL of each primer; and 1 μL of genomic DNA used as a template. The PCR reaction system is presented in Table 5. Previously collected positive samples for *A. phagocytophilum*, *A. bovis*, and *A. platys*, that were confirmed by sequencing, were used as positive controls. Every PCR reaction was carried out in the presence and absence of negative and positive controls. The amplified PCR products were further characterized and separated on the basis of 1.5% agarose gel electrophoresis and visualized using the gel imaging system UV light.

### 2.5. Sequencing and Phylogenetic Analysis

All *Anaplasma*-positive PCR products were subjected to capillary sequencing which was outsourced (Sangon Biotech Co., Shanghai, China). The retrieved sequences were edited, assembled, and trimmed using the DNAMAN 8.0 gene analysis software. We have used the BLAST algorithm, NCBI, for comparison with existing sequences in the GenBank database. The Clustal W program in MegAlign 7.2 (DNAStar, Madison, WI, USA) software was used to select representative sequences through multiple sequence alignment and to analyze the homology between the sequences obtained in the study and the known sequences. The obtained sequences were compared to check the difference among sequences using DNAMAN 8.0 gene analysis software and to build a phylogenetic tree using MEGA7.0. The phylogenetic tree was constructed using Maximum Likelihood (ML) algorithm, with 1000 Bootstrap value.

### 2.6. Statistical Analysis

A Chi square test using the SPSS software (version 23) was used to evaluate the correlation between the sex, age, breed, pesticide use, and tick infection of dogs and the infection rate of *Anaplasma* spp. *p* < 0.05 was considered statistically significant.

## 3. Results

### 3.1. Burden of Anaplasma spp. Infection in Hainan Province/Island

A total of 1051 dog blood samples from 18 cities and counties in Hainan Province (Table 6) were collected and examined. Species-specific infection rates for *Anaplasma* spp. which was detected in the present study are tabulated in Table 7. Overall, 6.0% (63/1051) of the samples were found to be positive for *A. platys*, 2.7% (28/1051) samples for *A. bovis*, and 1.0% (11/1051) samples were positive for *A. phagocytophilum*. *A. platys* was found to be widely distributed in Hainan Province. With the exception of a few cities like Dongfang City and Wuzhishan City, *A. platys* has been detected in every city of the Province. The infection rates of *A. platys* in Ledong, Tunchang, and Qiongzhong cities were found to be 43.75% (7/16), 53.33% (16/30), and 63.1% (12/19), respectively. Moreover, the distribution of *A. phagocytophilum* and *A. bovis* was found to be uneven. *A. phagocytophilum* was detected primarily in Haikou, Ding’an, Baoting, and Wanning, and *A. bovis* was detected in Haikou, Wenchang, Ding’an, Chengmai, Baoting, Lingao, and Sanya. It is worth noting that, in the 1051 dog blood samples that were sampled and processed further, we found three cases of mixed *Anaplasma* infection. One sample collected from Baoting area distinctively had a mixed infection of *A. bovis* and *A. platys.* In Ding’an city, two samples had mixed infection; one had *A. phagocytophilum* and *A. platys*, and the other had *A. bovis* and *A. platys* (Table 7).

### 3.2. Analysis of Relevant Risk Factors

The infection rate of *Anaplasma* spp. may be associated with some related risk factors, such as the breed, age, sex, feeding environment, usage of pesticides, contact of ticks, etc. [20]. In the process of blood collection in this experiment, we recorded details related to the sex, age, breeding type, pesticide use, and tick infection of each dog, and analyzed the influence of these factors on the infection rate of *Anaplasma* (Table 8). The results showed that, in terms of gender distribution, the infection rates of male and female dogs were found to be 7.54% and 11.15%, respectively. A Chi square analysis showed that gender had no statistical significance on the infection rate of *Anaplasma* spp. (X^2^ = 4.011, df = 1, *p* = 0.045). In terms of age distribution, the infection rate of *Anaplasma* spp. in younger dogs (1 year old) was found to be 12.61% which was higher than that in adult dogs (>1 year old) (7.78%). Statistical analysis showed that there was a correlation between age and the infection rate of *Anaplasma* (X^2^ = 6.430, df = 1, *p* = 0.011 *). We further found that the use of pesticides also contributed to the infection rate of *Anaplasma* (X^2^ = 4.939, df = 1, *p* = 0.026 *). The infection rate of *Anaplasma* in dogs using pesticides was 6.55% which was significantly lower than that in dogs not exposed to pesticides (11.27%). Moreover, the infection rate of *Anaplasma* was significantly affected by the presence of ticks on the body surfaces of the dogs (X^2^ = 47.933, df = 1, *p* < 0.0001 *). We found that the *Anaplasma* infection rate in dogs with ticks was 35.71%, which was significantly higher than that in dogs without ticks (7.94%); almost five times higher. The feeding environment of the dogs also had a significant impact on the infection rate of *Anaplasma* (X^2^ = 30.305, df = 2, *p* < 0.0001 *). The infection rate in rural dogs was 14.21% was higher than that in urban dogs (5.07%) and dogs in shelters (3.76%).

### 3.3. The Phylogenetic Analysis of the A. phagocytophilum and A. bovis 16S rRNA Gene

A total of 11 positive sequences of *A. phagocytophilum* and 28 positive sequences of *A. bovis* were obtained in this test. After comparing all the sequences, eight sequences were selected as representatives, which originated from Baoting: No. 8; Sanya: No. 2; Chengmai: No. 7; Ding’an: No. 4; No. 41, No. 61; Wenchang: No. 8; and Haikou: No. 8, and were named BT8 (OP793684); SY02 (OP788374); CM07 (OP788988); DA4 (OP788195); DA41 (OP788187); DA61 (OP788989); WC08 (OP788990); and HK8 (OP788371), respectively. Additionally, the known 16S rRNA partial sequences of *A. phagocytophilum* (registration numbers: GU724963, MN097866, KT944029, JN558813); *A. bovis* (KC311347, MF197898, KX450273, KU500914); *A. platys* (KU50914, MN227481); *A. marginale* (AF414875, ON528757); and *A. ovis* (KJ639880, AY262124) were used as reference sequences to construct phylogenetic trees (Figure 2). The results showed that DA41 was similar to *A. platys* (MN227481); DN4 was similar to *A. phagocytophilum* (KT944029); and BT8 and HK8 were similar to *A. phagocytophilum* (MN097866, JN558813) and fell in the same group. Furthermore, SY02, CM07, and DA61 were similar to *A. bovis* (KC311347, MF197898) and fell in the same group. The genetic distance between WC08 and *A. platys* (KU500914) was similar.

### 3.4. The Phylogenetic Analysis of A. platys gltA Gene

Sixty-three positive sequences of *A. platys* were retrieved in this study. After comparison of all sequences obtained, four strains were selected as representatives, and the known *gltA* gene sequence of *A. platys* (registration numbers: EU516387, AB058782, AY530807, KR011928) from Genbank was selected. Some exogenous sequence *A. marginale* (LC645238, MT722117), *A. ovis* (MG869297, MN238937), *A. centrale* (AF304141), *A. bovis* (KU586317), *A. phagocytophilum* (JQ622146), and *A. capra* (MH084720) were used as reference sequences to construct phylogenetic trees (Figure 3). The results show that all four positive sequences clustered in the same group.

## 4. Discussion

In the current study, we have investigated and analyzed the infection of *Anaplasma* spp. in dogs in Hainan Province/Island, and the total prevalence was found to be 9.7% (102/1051). Using approaches like conventional and nested PCR, we have screened the samples to check the presence of three species of *Anaplasma*—namely, *A. phagocytophilum*, *A. bovis*, and *A. platys*. The species and severity of the infection differed in different regions. Additionally, we found that samples collected from Dongfang and Wuzhishan Cities had no infection, which may be due to the small sample size. Samples collected from Wuzhishan City showed the absence of *Anaplasma* infection.

In the current study, the infection rate of *A. platys* was significantly higher than that of *A. phagocytophilum* and *A. bovis*. *A. platys* mainly leads to circulatory thrombocytopenia, which is highly prevalent in dogs [21]. A recent study has confirmed that 18.7% of dogs in the Caribbean Sea were found to be infected with *A. platys* [22]. Previous studies have also confirmed the existence of *A. platys* in dogs using PCR-based detection, in Brazil (16%) [23], Mexico (10%) [24], the United States (4.5%) [25], Portugal (75%) [26], Malaysia (3%) [27], and India (20%) [28]. In Southern China, Li et al. used RT-LAMP to detect *A. platys* in dog blood samples, with a positive rate of 62.1% (36/58) [29]. Our results showed that the total infection rates of *A. phagocytophilum* and *A. bovis* in dogs in Hainan were found to be 1% and 2.7%, respectively. In recent years, many such studies have reported the presence of *A. phagocytophilum* infection in dogs. In 2012, Zhang et al. reported that the average positive rate of *A. phagocytophilum* in dogs from 10 provinces and cities in China was about 10.05% [30]. Fukui et al. assessed the prevalence of *A. phagocytophilum* in 332 Japanese Ibaraki dogs in 2019. In yet another study, immunofluorescent antibody tests showed that 2.1% (7/328) of dog serum samples were positive for *A. phagocytophilum* [31].

The co-infection with multiple pathogens in dogs has also been commonly reported. In China, a recent study reported the triple infection case of *A. phagocytophilum*, *A. bovis,* and *A. platys* in dogs [32]. A double infection of *A. phagocytophilum* and *A. bovis* was found in ticks, cattle, and dogs in Japan [33,34]. Mixed infection of *Anaplasma* spp. in dogs was found in our study, including one case of mixed infection of *A. phagocytophilum* and *A. platys*, and two cases of mixed infection of *A. bovis* and *A. platys*. We have detected co-infection of *Anaplasma*, along with multiple tick-borne pathogens in dogs which were commonly detected in endemic areas. In a large retrospective serological study conducted in North America and the Caribbean, up to five vector-borne pathogens were detected in one dog [35]. In a kennel in North Carolina, serological evidence showed that 40% of the dogs were co-infected with *Anaplasma*, *Babesia canis*, *Babesia vensoni*, *Ehrlicha canis,* and *Rickettsia* [36]. In yet another study, 16.5% of American dog blood samples were found to be seropositive for more than one pathogen [37]. In Italy and Morocco, two serological investigations showed that 1.32% and 14.3% of dogs with dual seropositivity were detected, respectively [38,39]. Experimental studies on rodents show that co-infection regulates the host’s immune response to *A. phagocytophilum* and the production of interleukins (ILs), and reduces IFN-γ Level and the number of CD8 + T cells, leading to more serious clinical symptoms, increasing the burden of pathogens in blood and tissues, and leading to long-lasting infections [40,41,42,43]. This evidence indicates that the severity and complexity of clinical symptoms arising from mixed infection are enhanced and the possibility of disease is increased in mixed infections when compared to single species infections. *A. phagocytophilum* and *A. platys* are considered to be clinically significant tick-borne pathogens in dogs. Moreover, these two pathogens have the potential to threaten human health due to their zoonotic ability [44]. Serological and molecular evidence has suggested the presence of *A. phagocytophilum* infection in humans in the Americas, Asia, and Europe [45]. In addition, human granulocytic anaplasmosis caused by phagocytic anaplasmosis has been reported in Henan Province, China, which can lead to organ failure and death [46]. So far, much less is known about the pathogenicity of *A. bovis* in dogs. This pathogen has not been investigated much; therefore, there is a need for further in-depth study to clarify the relationship between the clinical symptoms and pathological results in dogs infected with *A. bovis*.

The age, breeding type, pesticide use, and tick infection of dogs are all correlated with the infection rate of *Anaplasma*. The infection rate of young dogs (12.61%) is significantly higher than that of adult dogs (7.78%), which may be due to their close contact with adult dogs with ticks, their inactivity, susceptibility to attachment to ticks, weak immunity, and susceptibility to tick-borne diseases. Compared to not using insecticides, the use of insecticides can reduce the infection rate of invertebrates, as these are diseases transmitted by external parasites and tick vectors. The use of insecticides can prevent dogs from being invaded by ticks from an external source. Tick infection has a very significant impact on the infection rate of *Anaplasma*. Surface infections of parasites—especially ticks—can promote the transmission of *Anaplasma* [47].

In this experiment, we divided all the dogs into three groups and found that the infection rate of rural dogs was higher than that of urban dogs and dogs in shelters. Most urban dogs come from families with higher incomes, and their owners regularly deworm and clean them to ensure that they are not disturbed by ticks. In addition, the behavior of city dogs will be restricted by their owners, and dogs are less likely to go into the grass or the wild to play, which reduces the chance of being infected with ticks. The dogs in the shelter are managed in a relatively standardized manner and regularly deworming, so their infection rate of *Anaplasma* is low. Most rural dogs are local breeds, and these local breeds come from low socio-economic backgrounds, with low opportunities for vaccination and deworming. Moreover, some dogs will be freely released and will have more opportunities to go to lush areas where ticks often appear [48]. In addition, some captive dog houses in rural areas are often located in remote corners, surrounded by dense trees and grass, and the breeding environment is not very good. These conditions can easily lead to dogs being infected by ticks and suffering from tick-borne diseases.

According to the current research, the impact of gender on the infection rate of *Anaplasma* is not yet clear. The results of this experiment show that gender has no statistically significant impact on the infection rate of *Anaplasma*. However, in the investigation of *Anaplasma* and tick-borne parasites in dogs in Malawi, it was found that the probability of male dogs (54.8%) being infected with tick-borne diseases is higher than that of female dogs (45.2%) (X^2^ = 5.3512, df = 1, *p* = 0.020708). The main reason is that most male dogs in Malawi usually live in groups during the breeding season, as fighting for females is common, which increases their chances of being infested with ticks [49]. This research result is also consistent with the findings of neighboring Zambia [50]. However, this contrasts sharply with other previous studies. The research reports of Galay et al. [48] and Konto et al. [51] indicate that the incidence of ticks in female dogs is higher.

## 5. Conclusions

In the present study, we have tried to excavate and explore the prevalence of *Anaplasma* spp. infections in dogs from 18 cities/counties of Hainan Province/Island. Our results showed that there were three species of *Anaplasma* detected from whole blood samples of dogs; namely, *A. phagocytophilum* (1.0%), *A. bovis* (2.7%), and *A. platys* (6.0%). The overall prevalence of *Anaplasma* is 9.7% (102/1051). The analysis of relevant risk factors showed that the age, breeding type, pesticide use, and tick infection of dogs are significant risk factors.

These findings reveal the transmission factors of *Anaplasma* spp. to a certain extent and provide significant support and information to the epidemic prevention and control bodies of the Province. They also highlight and provide a road map for the strategic control and management of the infection and disease in a well-structured manner.

## Figures and Tables

**Figure 1 vetsci-10-00339-f001:**
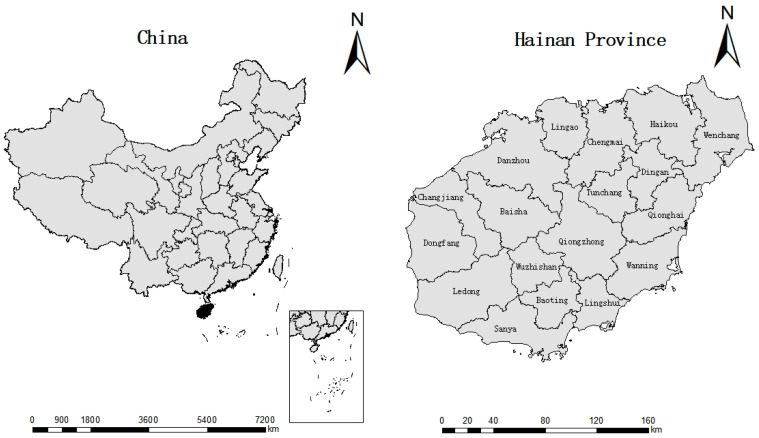
The geographical location of the study area (Hainan Province, China) and the distribution sites of different cities and counties.

**Figure 2 vetsci-10-00339-f002:**
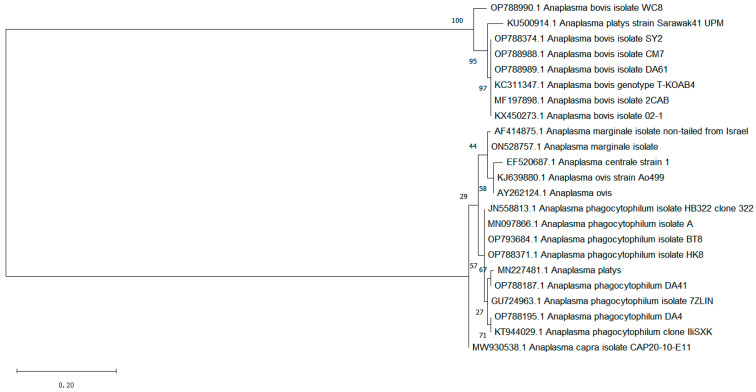
The phylogenetic analysis of the *A. phagocytophilum* and *A. bovis* 16S rRNA gene sequences by the Maximum Likelihood method. The evolutionary history was inferred by using the Maximum Likelihood method based on the General Time Reversible model. The tree with the highest log likelihood (−605.10) is shown. The percentage of trees in which the associated taxa clustered together is shown next to the branches. Initial tree(s) for the heuristic search were obtained automatically by applying Neighbor-Join and BioNJ algorithms to a matrix of pairwise distances estimated using the Maximum Composite Likelihood (MCL) approach, and then selecting the topology with superior log likelihood value. A discrete Gamma distribution was used to model evolutionary rate differences among sites (5 categories (+G, parameter = 0.9458)). The tree is drawn to scale, with branch lengths measured in the number of substitutions per site. The analysis involved 23 nucleotide sequences. Codon positions included were 1st + 2nd + 3rd + Noncoding. All positions containing gaps and missing data were eliminated. There were a total of 162 positions in the final dataset. Evolutionary analyses were conducted in MEGA7.

**Figure 3 vetsci-10-00339-f003:**
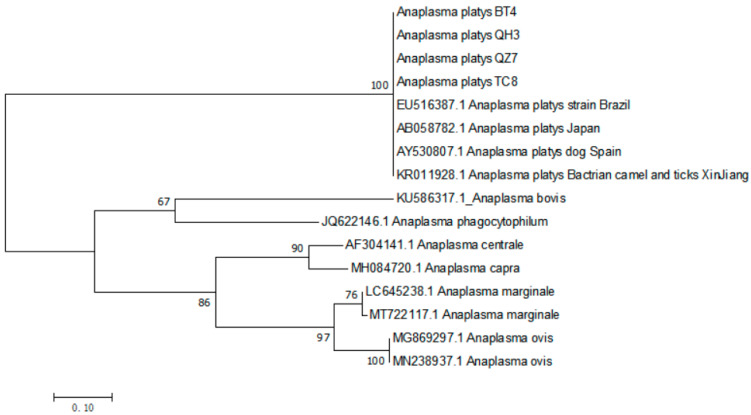
The phylogenetic analysis of *A. platys gltA* gene sequence by the Maximum Likelihood method. The evolutionary history was inferred by using the Maximum Likelihood method based on the Tamura–Nei model. The tree with the highest log likelihood (−1693.81) is shown. The percentage of trees in which the associated taxa clustered together is shown next to the branches. Initial tree(s) for the heuristic search were obtained automatically by applying Neighbor-Join and BioNJ algorithms to a matrix of pairwise distances estimated using the Maximum Composite Likelihood (MCL) approach, and then selecting the topology with superior log likelihood value. A discrete Gamma distribution was used to model evolutionary rate differences among sites (5 categories (+G, parameter = 1.4671)). The tree is drawn to scale, with branch lengths measured in the number of substitutions per site. The analysis involved 16 nucleotide sequences. Codon positions included were 1st + 2nd + 3rd + Noncoding. All positions containing gaps and missing data were eliminated. There were a total of 282 positions in the final dataset. Evolutionary analyses were conducted in MEGA7.

**Table 1 vetsci-10-00339-t001:** Information of blood samples collected from dogs in Hainan Province, China.

Location	Number of Samples	Location	Number of Samples
Haikou	636	Lingshui	4
Qiongzhong	19	Baisha	14
Dongfang	2	Chenmai	16
Wenchang	31	Baoting	24
Changjiang	15	Wanning	18
Danzhou	36	Ledong	16
Dingan	107	Tunchang	30
Wuzhishan	12	Lingao	5
Qionghai	34	Sanya	32

**Table 2 vetsci-10-00339-t002:** Primer sequences for *Anaplasma*.

Target Gene	Primer	Sequence 5′ to 3′	Amplicon Size (bp)	References
16S rRNA	EE-1	TCCTGGCTCAGAACGAACGCTGGCGGC	1433	Barlough et al. [17]
	EE-2	AGTCACTGACCCAACCTTAAATGGCTG		
16S rRNA	SSAP2f	GCTGAATGTGGGGATAATTTAT	641	Kawahara et al. [18]
	SSAP2r	ATGGCTGCTTCCTTTCGGTTA		
16S rRNA	AB1f	CTCGTAGCTTGCTATGAGAAC	551	
	AB1r	TCTCCCGGACTCCAGTCTG		
*gltA*	*gltAf*	GACCTACGATCCGGGATTCA	580	Silva et al. [19]
	*gltAr*	CCGCACGGTCGCTGTT		

**Table 3 vetsci-10-00339-t003:** PCR amplification conditions of *Anaplasma*.

Category	Primer	Amplification Conditions					
*A. bovis*	EE1 EE2	94 °C	94 °C	55 °C	72 °C	35	72 °C
		5 min	30 s	30 s	30 s		5 min
	AB1f AB1r	94 °C	94 °C	58 °C	72 °C	40	72 °C
		5 min	30 s	30 s	30 s		10 min
*phagocytophilum*	EE1 EE2	94 °C	94 °C	55 °C	72 °C	35	72 °C
		5 min	30 s	30 s	30 s		5 min
	SSAP2f SSAP2r	94 °C	94 °C	58 °C	72 °C	40	72 °C
		5 min	30 s	30 s	30 s		10 min
*A. platys*	*gltAF* *gltAR*	94 °C	94 °C	60 °C	72 °C	35	72 °C
		3 min	1 min	1 min	1 min		5 min

**Table 4 vetsci-10-00339-t004:** nPCR reaction system.

First		Second	
Components	Volume (μL)	Components	Volume (μL)
2 × Taq Plus Master Mix II (Dye Plus)	12.5	2 × Taq Plus Master Mix II (Dye Plus)	12.5
Forward Primer	0.5	Forward Primer	0.7
Reverse Primer	0.5	Reverse Primer	0.7
Template	1	Template	1
ddH_2_O	10.5	ddH_2_O	10.1
Total	25	Total	25

**Table 5 vetsci-10-00339-t005:** PCR reaction system.

Components	Volume (μL)
2 × Taq Plus Master Mix II (Dye Plus)	12.5
Forward Primer	0.5
Reverse Primer	0.5
Template	1
ddH_2_O	10.5
Total	25

**Table 6 vetsci-10-00339-t006:** Anaplasmosis infection in dogs in different areas.

Location	No. Tested	Positive	No. Infected/(%)
Haikou	636	16	2.5
Qiongzhong	19	12	63.2
Dongfang	2	0	0
Wenchang	31	7	22.6
Changjing	15	1	6.7
Danzhou	36	7	19.4
Dingan	107	12	11.2
Wuzhishan	12	0	0
Qionghai	34	5	14.7
Lingshui	4	1	25
Baisha	14	1	7.1
Chengmai	16	2	12.5
Baoting	24	6	25
Wanning	18	1	5.5
Ledong	16	7	43.8
Tunchang	30	16	53.3
Lingao	5	4	80
Sanya	32	1	3.1
Total	1051	99	9.4

**Table 7 vetsci-10-00339-t007:** Infection of different species of *Anaplasma*.

Location	No. Tested	No. Infected/(%)		
		*A. phagocytophilum* (%)	*A. bovis* (%)	*A. platys* (%)
Haikou	636	0.15(1/636)	2.20(14/636)	0.15(1/636)
Qiongzhong	19	0(0/19)	0(0/19)	63.16(12/19)
Dongfang	2	0(0/2)	0(0/2)	0(0/2)
Wenchang	31	0(0/31)	16.13(5/31)	6.45(2/31)
Changjing	15	0(0/15)	0(0/15)	6.67(1/15)
Danzhou	36	0(0/36)	0(0/36)	19.44(7/36)
Dingan	107	5.61(6/107)	1.87(2/107)	5.61(6/107)
Wuzhishan	12	0(0/12)	0(0/12)	0(0/12)
Qionghai	34	0(0/34)	0(0/34)	14.71(5/34)
Lingshui	4	0(0/4)	0(0/4)	25.0(1/4)
Baisha	14	0(0/14)	0(0/14)	7.14(1/14)
Chengmai	16	0(0/16)	5.26(1/16)	6.25(1/16)
Baoting	24	12.50(3/24)	4.17(1/24)	12.50(3/24)
Wanning	18	5.55(1/18)	0(0/18)	0(0/18)
Ledong	16	0(0/16)	0(0/16)	43.75(7/16)
Tunchang	30	0(0/30)	0(0/30)	53.33(16/30)
Lingao	5	0(0/5)	80.0(4/5)	(0/5)
Sanya	32	0(0/32)	3.13(1/32)	0(0/32)
Total	1051	1.0(11/1051)	2.7(28/1051)	6.0(63/1051)

**Table 8 vetsci-10-00339-t008:** Influence of different factors on infection rate of *Anaplasma*.

Factor	Parametrs	Number	Infection Rate (%)	Statistical Analysis
Gender	Male	547	11.15(61/547)	X^2^ = 4.011, df = 1, *p* = 0.045
	Female	504	7.54(38/504)	
Age	<1 year	357	12.61(45/357)	X^2^ = 6.430, df = 1, *p* = 0.011 *
	>1 year	694	7.78(54/694)	
Pesticide	Yes	412	6.55(27/412	X^2^ = 4.939, df = 1, *p* = 0.026 *
	No	639	11.27(72/639)	
Tick infection	Yes	56	35.71(20/56)	X^2^ = 47.933, df = 1, *p* < 0.0001 *
	No	995	7.94(79/995)	
Feeding mode	urban dogs	217	5.07(11/217)	X^2^ = 30.305, df = 2, *p* < 0.0001 *
	dogs in shelters	292	3.76(11/292)	
	rural dogs	542	14.21(77/542)	

*: 5% Significance level

## Data Availability

All the sequences used in this article will be uploaded in GenBank when the manuscript is accepted for publication. The data that support the findings of this study are available from the corresponding authors upon reasonable request. The extracted DNA of the blood samples will be made available upon request in case there is leftover material.

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
