# Peer review of "Molecular Detection and Phylogenetic Characterization of *Anaplasma* spp. in Dogs from Hainan Province/Island, China"

_vetsci, 2023, doi:10.3390/vetsci10050339_

Round 1
Reviewer 1 Report
Dear editor and authors, the manuscript deals with Molecular detection and phylogenetic characterization of Anaplasma spp. in dogs from China. I recommend publishing the manuscript in the form it was presented. I ask that the authors pay attention to highlighting in italics all the genera and species mentioned in the manuscript, as recommended by the international code of zoological nomenclature.
1. Did you detect plagiarism? Do you have any other ethical concerns about this study?
No
2. What is the main question addressed by the research? Is it relevant and interesting?
In the MS the authors detected Anaplasma spp. in dogs from China. It is relevant once the authors carried out the molecular characterization of the species and are contributing to the knowledge of this very complex group. It is very relevant.
3. How original is the topic? What does it add to the subject area compared with other published material?
Average, the results include new Anaplasma sequences that may be used in the future for distribution studies and species validation.
4. Is the paper well written? Is the text clear and easy to read?
Yes
5. Are the conclusions consistent with the evidence and arguments presented? Do they address the main question posed?
Yes
6. Could you please provide more detailed/specific suggestions or grammar corrections in order to better help to improve the quality of the paper?
The language is appropriate and the authors have to revise the names of the species that must be written in italics, as suggested by the international code of zoological nomenclature.
kind regards,
Author Response
Point
I ask that the authors pay attention to highlighting in italics all the genera and species mentioned in the manuscript, as recommended by the international code of zoological nomenclature.
Response
We appreciate it very much for this good suggestion, and we have done it according to your ideas. All genera and species mentioned in the manuscript have been highlighted in italics in accordance with the International zoology Nomenclature.
Reviewer 2 Report
The currently available manuscript, "Molecular detection and phylogenetic characterization of Anaplasma spp. in dogs from Hainan province/island, China" is interesting and overall well written. However, some modifications are necessary before publication. Some of these points are mentioned below. Additionally, the authors should be careful with scientific names; there are several names that are misspelled, and throughout the manuscript, many names are not written in italics. Please check this throughout the manuscript.
Abstract:
Please add the overall prevalence of Anaplasma
Change A. phagocytophum to A. phagocytopihum and A. bovies to A. bovis.
Add a dot after Anaplasma spp
Introduction
Please add a paragraph, including information related to the epidemiology of anaplasmosis in dogs as well as some risk factors previously identified. These informations are important to the current study.
Material and methods
2.4. PCR amplification
Why did the authors use a 16S PCR reaction for A. phagocytophilum and A. bovis and not a broad-range PCR assay for amplification of Anaplasma spp.? Using this approach, the positive samples could be submitted to additional PCR targeting other genes for a better molecular characterization.
What happened with the groEL PCR to detect A. platys?
Tables 2, 3, and 4 should be summarized in only one table. The authors must keep in the text only the essential information for understanding the methods and include the citations of the papers where the protocols were originally published.
2.5. Sequencing and phylogenetic analysis
What software was used for phylogenetic analysis?
What evolutionary model was used in the phylogenetic analysis, and how was it selected?
Why did the authors use the NJ analysis instead of maximum likelihood (ML) or Bayesian inference (BI)? Compared to NJ, the ML and BI are more efficient.
Results
I strongly suggest the authors perform a robust molecular characterization using additional genes. In this way, all the positive samples, or at least the representative sequences, may be analyzed using other genes (groEL, gltA, etc.) and not using only one target as herein presented.
From my point of view, the sequences of different species (A. phagocitophylum, A. bovis, and A. platys) from the same gene should be analyzed in the same phylogenetic tree. Also, the analysis must include other different Anaplasma species and not only the one herein analyzed.
3.1. Burden of Anaplasma spp. infection in Hainan province/island.
This section may be summarized once several pieces of information described in the text are presented in tables 6 and 7.
3.2. Analysis of relevant risk factors
"The infection rate of Anaplasma spp. may be associated with some related risk factors, such as the breed, age, sex, feeding environment, usage of pesticides, contact of ticks, etc". Please add a reference for this paragraph.
"The feeding environment of dogs". What does it mean? Please clarify.
4. Discussion
In this section, the authors do not discuss the different factors (age, ticks, etc) that affect the infection rate of Anaplasma. How could these differences be explained? This topic must be addressed in this section.
"However, Anaplasma pathogen was detected in the goat and cattle blood samples that were collected from Wuzhishan City." Please add a reference for this sentence.
"In this study, A. bovis was detected in dogs with no clinical symptoms." In the M&M section, the authors did not mention that the dogs were submitted to a clinical examination, only a physical examination to check for the presence of ticks. Thus, the authors must clarify if all dogs were clinically examined and if the positive animals showed clinical signs (e.g., fever, lethargy, or inappetence).
5. Conclusions
This section must be rewritten. There are too many items that were previously mentioned in the results or in the introduction sections, that may be deleted. The authors could be more concise and only present the most important findings.
Author Response
Point 1: Abstract:Please add the overall prevalence of Anaplasma Change A. phagocytophum to A. phagocytopihum and A. bovies to A. bovis. Add a dot after Anaplasma spp
Response 1: The overall prevalence of Anaplasma has been added to the abstract. All text errors have been modified.
Point 2: Introduction Please add a paragraph, including information related to the epidemiology of anaplasmosis in dogs as well as some risk factors previously identified. These informations are important to the current study.
Response 2: Thank you to the reviewer for this excellent suggestion. We have added this information in the introduction.
Point 3: Material and methods
2.4. PCR amplification
Why did the authors use a 16S PCR reaction for A. phagocytophilum and A. bovis and not a broad-range PCR assay for amplification of Anaplasma spp.? Using this approach, the positive samples could be submitted to additional PCR targeting other genes for a better molecular characterization.
Response 3: At present, because 16S rRNA gene is highly conservative, PCR detection of Anaplasma spp. based on 16S rRNA gene is the most common.
Point 4: What happened with the groEL PCR to detect A. platys?
Response 4: I’m so sorry, this is a writing error. The gltA gene was used to detect A. platys. We are very sorry for our careless mistake and it was rectified.
Point 5: Tables 2, 3, and 4 should be summarized in only one table. The authors must keep in the text only the essential information for understanding the methods and include the citations of the papers where the protocols were originally published.
Response 5: In our opinion, three tables can provide more detailed data, and combining them into one table would appear cumbersome. But if you think this is necessary, we will correct it in the next revision.
Thanks for reviewer’s comments.
Point 6: Material and methods
2.5. Sequencing and phylogenetic analysis
What software was used for phylogenetic analysis? What evolutionary model was used in the phylogenetic analysis, and how was it selected? Why did the authors use the NJ analysis instead of maximum likelihood (ML) or Bayesian inference (BI)? Compared to NJ, the ML and BI are more efficient.
Response 6: MEGA is used for phylogenetic analysis. We use Neighbor-Joining (NJ) because of its fast advantage and use the default evolutionary model. If you think this is necessary, we will change the method in the next repair.
Point 7: Results
I strongly suggest the authors perform a robust molecular characterization using additional genes. In this way, all the positive samples, or at least the representative sequences, may be analyzed using other genes (groEL, gltA, etc.) and not using only one target as herein presented.
Response 7:
Thank you very much for the reviewer's suggestion, but reusing the new gene target would be a significant workload for us, which is unacceptable. We will adopt your suggestion in future experiments.
Point 8: From my point of view, the sequences of different species (A. phagocitophylum, A. bovis, and A. platys) from the same gene should be analyzed in the same phylogenetic tree. Also, the analysis must include other different Anaplasma species and not only the one herein analyzed.
Response 8: We only explored the genetic relationships between the same species in different regions. Your suggestion is very meaningful, and we will adopt it in future research.
Point 9: 3.2. Analysis of relevant risk factors
"The infection rate of Anaplasma spp. may be associated with some related risk factors, such as the breed, age, sex, feeding environment, usage of pesticides, contact of ticks, etc". Please add a reference for this paragraph.
Response 9: Relevant references have been added.
Point 10: "The feeding environment of dogs". What does it mean? Please clarify.
Response 10: Some dogs are kept in captivity, while others are kept in the wild.
Point 11: 4. Discussion
In this section, the authors do not discuss the different factors (age, ticks, etc) that affect the infection rate of Anaplasma. How could these differences be explained? This topic must be addressed in this section.
Response 11: We appreciate it very much for this good suggestion, and we have done it according to your ideas. We have added relevant content in the discussion section of the manuscript.
Point 12: "However, Anaplasma pathogen was detected in the goat and cattle blood samples that were collected from Wuzhishan City." Please add a reference for this sentence.
Response 12: This is our other survey, and the data has not been published yet. Therefore, we have decided to delete this sentence.
Point 13: "In this study, A. bovis was detected in dogs with no clinical symptoms." In the M&M section, the authors did not mention that the dogs were submitted to a clinical examination, only a physical examination to check for the presence of ticks. Thus, the authors must clarify if all dogs were clinically examined and if the positive animals showed clinical signs (e.g., fever, lethargy, or inappetence).
Response 13: We cannot conduct long-term observations during sampling, so we can only perform simple surface examinations, and if some clinical symptoms have not yet been manifested, they cannot be observed. Therefore, we are unable to contact the previous text and have decided to delete this sentence.
Point 14: 5. Conclusions
This section must be rewritten. There are too many items that were previously mentioned in the results or in the introduction sections, that may be deleted. The authors could be more concise and only present the most important findings.
Response 15: The conclusion has been rewritten.
Reviewer 3 Report
Comments on the article “Molecular detection and phylogenetic characterization of Anaplasma spp. in dogs from Hainan province/island, China” by Lin and colleagues - Manuscript vetsci-2347674.
The study aims to examine Anaplama in dogs in the Hainan Province/Island using molecular techniques that may be able to detect such pathogen in animals also exhibiting subclinical symptoms. The introduction contains all the information required to understand why the authors proposed to examine this pathogen. The experimental design is solid, and the text is well-written regarding English.
I have no major remarks to make, but only a few minor pointers and considerations for the authors.
· Anaplasma indicates a genus, so it must be written in italics. Correct throughout the text, including figures, tables, and references.
· A. phagocytophilum, A. marginale, A. centrale, A. ovis, A. bovis,…… Anaplasma phagocytophilum, Anaplasma marginale, Anaplasma centrale,……indicate genus and species and must be written in italics. Correct throughout the text, including figures and tables, and check for any similar errors for any genus e species, also in the references.
· Sometimes the authors write species, other times spp, and spp with a dot (spp.). Choose a style and match the text.
· Why were the animal populations quite variable
and sometimes excessively low?
The presence of ticks is not taken into account during the experiments.
· The authors merged different samplings into one study. This could make the description of the results somewhat fragmented.

Author Response
Point 1: Anaplasma indicates a genus, so it must be written in italics. Correct throughout the text, including figures, tables, and references. A. phagocytophilum, A. marginale, A. centrale, A. ovis, A. bovis,……Anaplasma phagocytophilum, Anaplasma marginale, Anaplasma centrale,……indicate genus and species and must be written in italics. Correct throughout the text, including figures and tables, and check for any similar errors for any genus e species, also in the references.
Response 1: We appreciate it very much for this good suggestion, and we have done it according to your ideas. All genera and species mentioned in the manuscript have been highlighted in italics in accordance with the International zoology Nomenclature.
Point 2: Sometimes the authors write species, other times spp, and spp with a dot (spp.). Choose a style and match the text.
Response 2: Thank you very much for the suggestion. All texts have been unified.
Point 3: Why were the animal populations quite variable and sometimes excessively low?
Response 3: As you said, the samples we collected from each place were not uniform, because the impact of the COVID-19 epidemic made it difficult for us to leave the city to collect samples from other places.
Point 4: The presence of ticks is not taken into account during the experiments.
Response 4: The influencing factors of ticks on the experiment are located in Results 3.2 of the manuscript.
Point 5: The authors merged different samplings into one study. This could make the description of the results somewhat fragmented.
Response 5:Well, thank you very much for the reviewer's suggestions on this manuscript, but I'm sorry I didn't quite understand the meaning of this sentence. The samples in our manuscript are all for dog blood and there are not many different samples. Thank you again for taking the valuable time to provide suggestions for this manuscript.
Round 2
Reviewer 2 Report
Dear authors
The manuscript has been improved. However, the modifications related to phylogenetic analysis were not addressed. From my point of view, these changes are important and can bring even more robustness to the results. Therefore, I suggest the authors rerun the phylogenetic analysis using the previous suggestions.
Author Response
We have conducted a new phylogenetic analysis based on your suggestion and revised the relevant results in the manuscript.